# Immunogenic Cell Death Enhances Immunotherapy of Diffuse Intrinsic Pontine Glioma: From Preclinical to Clinical Studies

**DOI:** 10.3390/pharmaceutics14091762

**Published:** 2022-08-24

**Authors:** Guohao Liu, Yanmei Qiu, Po Zhang, Zirong Chen, Sui Chen, Weida Huang, Baofeng Wang, Xingjiang Yu, Dongsheng Guo

**Affiliations:** 1Department of Neurosurgery, Tongji Hospital, Tongji Medical College, Huazhong University of Science and Technology, Wuhan 430000, China; 2Department of Neurology, Union Hospital, Tongji Medical College, Huazhong University of Science and Technology, Wuhan 430000, China; 3Department of Neurosurgery, Union Hospital, Tongji Medical College, Huazhong University of Science and Technology, Wuhan 430000, China; 4Department of General Surgery, Jinshan Hospital, Fudan University, Shanghai 200433, China; 5Department of Histology and Embryology, School of Basic Medicine, Tongji Medical College, Huazhong University of Science and Technology, Wuhan 430000, China

**Keywords:** diffuse intrinsic pontine glioma, immune microenvironment, immunotherapy, immunogenic cell death, damage associated molecular patterns

## Abstract

Diffuse intrinsic pontine glioma (DIPG) is the most lethal tumor involving the pediatric central nervous system. The median survival of children that are diagnosed with DIPG is only 9 to 11 months. More than 200 clinical trials have failed to increase the survival outcomes using conventional cytotoxic or myeloablative chemotherapy. Immunotherapy presents exciting therapeutic opportunities against DIPG that is characterized by unique and heterogeneous features. However, the non-inflammatory DIPG microenvironment greatly limits the role of immunotherapy in DIPG. Encouragingly, the induction of immunogenic cell death, accompanied by the release of damage-associated molecular patterns (DAMPs) shows satisfactory efficacy of immune stimulation and antitumor strategies. This review dwells on the dilemma and advances in immunotherapy for DIPG, and the potential efficacy of immunogenic cell death (ICD) in the immunotherapy of DIPG.

## 1. Introduction

Diffuse intrinsic pontine glioma (DIPG) is the most lethal tumor involving the pediatric central nervous system [1]. DIPG originates in the pons and is characterized by diffuse infiltration and poor demarcation from normal tissue, and frequent invasion of distant brain regions [2]. Due to its delicate anatomical location, it is impossible to achieve significant resection [3]. Despite nearly five decades of research, the median survival of children that are diagnosed with DIPG is only 9 to 11 months [4]. Palliative radiotherapy remains the only clinically effective treatment option, although with only a three-month survival benefit [5]. Hence, it is imperative to develop new therapeutic strategies to alleviate or cure this malignant and fatal disease.

In recent years, tumors that are associated with the pediatric central nervous system have surpassed hematological tumors as the leading cause of cancer-related deaths in children and adolescents [6]. It may be largely due to the widespread use of immunotherapy in hematological malignancies. Immunotherapy is considered to be a milestone in precision medicine. Advances in immunotherapy have significantly improved the prognosis of patients that are diagnosed with cancer [7]. Most critically, it elicits a dramatic therapeutic response in patients resistant to other conventional treatments [8]. Immunotherapy strategies, such as the use of immune checkpoint inhibitors (ICIs), chimeric antigen receptor (CAR) T-cell therapy, vaccine therapy, and oncolytic viruses present exciting opportunities to cure DIPG with a unique and heterogeneous feature. Nevertheless, inadequate knowledge of the DIPG immune microenvironment hinders the use of immunotherapeutic modalities.

The current consensus is that DIPG is a cold tumor, indicating that DIPG shows limited immune cell infiltration, reduced secretion of inflammatory factors, rare antigen-presenting cells, isolated and defective immune killing mechanisms, and a nearly deserted immune microenvironment [9,10]. The non-inflammatory DIPG microenvironment greatly limits the role of immunotherapy in DIPG. For instance, patients with DIPG do not experience any survival benefit when they are treated with ICIs [11,12,13]. The immunosuppressive effect of steroids and the compactness of the blood-brain barrier (BBB) are also important features hindering its application [14]. Besides, toxicity that is associated with immunotherapy is a challenge [15]. Therefore, it is imperative to develop a better theoretical basis and technical tools to address the limitations of immunotherapy in DIPG.

Immunogenic cell death (ICD) is a form of programed cell death that is induced by multiple antitumor therapies. It is accompanied by the release of damage-associated molecular patterns (DAMPs), which facilitate the maturation of dendritic cells (DCs) and tumor death that is induced by cytotoxic T lymphocytes (CTLs) [16]. Intra-tumoral delivery of vaccines, oncolytic viruses, CAR-T-cells, or chemotherapeutics inducing the release of DAMPs may be a promising therapeutic strategy, suggesting that the assessment of ICD-related molecules in serum and cerebrospinal fluid may have potential diagnostic significance [17]. Hence, ICD may be a promising method contributing to the immunotherapeutic efficacy.

In this review, we summarize the current studies involving the non-inflammatory DIPG immune microenvironment and immunotherapy strategies targeting DIPG, and address the dilemma of immunotherapy for DIPG. We also emphasize the potential role of ICD in enhancing the efficacy of immunotherapy for DIPG. Our review provides a new perspective on personalized medicine and precision treatment of DIPG.

## 2. Diagnosis, Current Management, and Molecular Characteristics of DIPG

The diagnosis of DIPG is based on clinical manifestations such as ataxia, pyramidal tract dysfunction, and abducens nerve (cranial nerve VI) palsy [18]. Magnetic resonance imaging (MRI) is still the most powerful modality for the clinical diagnosis of DIPG. MRI of patients with DIPG shows a space-occupying lesion in the pontine region, which is relatively hypointense or isointense on T1-weighted images and hyperintense on T2-weighted images, compared with a normal brain [19] (Figure 1). Palliative local radiotherapy remains the only standard treatment option for DIPG, and in the absence of radiotherapy, the overall survival is only 5 months [5]. Although temozolomide showed some efficacy in adult high-grade glioma, it does not improve the prognosis of DIPG [20]. Thus, palliative care services are indispensable for children with DIPG and their families, including individualized physical, psychological, social, and other forms of support early in the disease [21].

The therapeutic modalities for DIPG have traditionally been based on the molecular landscape of adult gliomas, with similar biological behaviors and molecular patterns. Over the past three decades, more than 250 clinical trials have shown no significant improvement in DIPG treatment [22,23]. Studies have established primary DIPG cells in vitro as well as orthotopic animal models of intracranial DIPG tumorigenesis [24,25,26,27,28] using improved biopsy techniques [29,30,31]. The rapid development of high-throughput technology and proteomics has provided a relatively unambiguous molecular landscape of DIPG [28,32,33,34]. DIPG is associated with a unique pathological behavior and molecular pattern compared with non-midline pediatric high-grade glioma (pHGG) and adult HGG (aHGG) [35,36].

The mutations that were found in H3K27M were hailed as the most important discovery in DIPG [37,38]. H3K27M, a somatic mutation of histone H3 involving a substitution of lysine by methionine at position 27, occurs primarily in the genes encoding histones H3.1 and H3.3, HIST1H3B, and H3F3A, respectively [38,39,40,41]. H3K27M is found in approximately 80% of patients with DIPG (H3.3K27M: 70% of DIPG; H3.1K27M: 15% of DIPG) and is associated with a range of characteristic clinical manifestations and worse treatment tolerance, and represents unsatisfactory prognosis and clinical outcomes (H3.1K27M, median overall survival: 15 months; H3.3K27M, median overall survival: 11 months) [41]. Therefore, the mutation was newly labeled as “diffuse midline glioma H3K27M-mutant” in the 2016 World Health Organization (WHO) tumor classification [42] and the 2021 WHO tumor classification [43]. Further, H3K27M is closely related to other mutations in DIPG, such as H3.1K27M-related ACVR1 mutation (25% of DIPG) and H3.3K27M-related amplified mutations in CCND2, PDGFRA (30% of DIPG), TP53 (75% of DIPG), and MYC [36,44,45,46]. In addition, RB phosphorylation (30% of DIPG), as well as STAT3 and PPM1D amplification (9–23% of DIPG), were identified in patients with DIPG [47,48,49] (Table 1).

The discovery of the unique molecular landscape provides an important theoretical basis for the development of DIPG-targeted drugs [35,50]. Currently, several classes of drugs have been used in pre-clinical experiments and clinical trials to treat DIPG, such as histone deacetylase and demethylase inhibitors, H3K27 demethylase inhibitor, Zeste enhancer homologue-2 (EZH2) inhibitors, DNA methylation inhibitors, and inhibitors of bromodomain and extra-terminal motif proteins [51,52,53,54,55,56,57,58,59,60].

## 3. Immune Microenvironment of DIPG

Currently, the immune microenvironment of DIPG remains unclear due to the difficulty of obtaining tumor samples and the relevance of intracranial animal models [73]. However, the tumor immune microenvironment differs greatly between pediatric and adult gliomas [74]. Further, DIPG exhibits a unique immune microenvironment compared with non-midline pediatric gliomas and adult glioblastoma (GBM) (Figure 2) [10].

### 3.1. Immunological Subgroups of DIPG

Vinci et al. reviewed subclonal genomic analyses to report a remarkable intertumoral heterogeneity of DIPG [75]. A previous study identified six immune subtypes of cancer tissue (C1, wound healing; C2, IFN-gamma dominant; C3, inflammatory; C4, lymphocyte depleted; C5, immunologically quiet; C6, TGF-β dominant), in an effort to determine immunogenicity across cancer types [76]. Accordingly, Zhu et al. classified patients with DIPG into three immune subgroups: lymphocyte-depleted (50%), immunologically quiet (14%), and an inflammatory subtype (14%) [77]. In this study, nearly 80% of patients with DIPG carried a “barren” and “silent” immune status of unresponsiveness, although the inflammatory subgroup exhibited a prolonged survival and a favorable response to irradiation.

### 3.2. Tumor-Associated Macrophages (TAMs)

Tumor-associated macrophages (TAMs) are the predominant subtype of immune cells infiltrating pediatric and adult gliomas [78]. TAMs sustain tumor proliferation and mediate immunosuppression [79,80]. Further, they facilitate the progression of cerebral edema, and induce GBM resistance to chemoradiotherapy [81,82]. In adult GBM, the M1-like and M2-like phenotypic classification of TAM is still widely used, despite controversy [83,84]. However, DIPG-associated macrophages do not fully fit the M1 or M2 classification [9]. The recruitment of M1-like macrophages in Sonic Hedgehog (SHH) medulloblastomas correlates with poor prognosis, suggesting the unique biological behavior of TAMs in pediatric CNS tumors [85].

Transcriptome sequencing and flow cytometry of DIPG tissues demonstrated that M1-like macrophages were not significantly different in the DIPG immune microenvironment compared with normal brain tissues; however, the number of CD45^+^Iba1^+^ microglia were increased [9]. And the proportion of macrophages in CD45^+^ leukocytes was 95% compared with 70% in adult GBM [9]. Besides, H3.3-K27M DIPG cells that were co-cultured with macrophages in vitro manifested little effect on macrophage phenotype compared with U87 cells co-cultures [10]. It is interesting to explore the discrepancy in the activation of microglia/macrophage between DIPG and adult GBM.

### 3.3. Tumor-Associated Lymphocytes

Tumor-infiltrating lymphocytes (TILs) play a pivotal role in the tumor microenvironment. Not only the amount of the infiltrating TILs, but the subtype also determines the clinical outcome and therapeutic effect of patients with tumors [86,87,88]. Type 1 T-cells, such as CD4 T-helper 1 (Th1), facilitate antigen presentation, whereas CD8 cytotoxic T-cells (CTL) accelerate tumor destruction [89]. Conversely, Type 2 CD4 T-helper cells (Th2), such as Tregs (FOXP3 + CD4 regulatory T-cells) functionally reduce the number of CTL to enhance tumor growth, resulting in an immunosuppressive microenvironment [90].

Previous studies demonstrated that the increased infiltration of CD8+ T-cells in adult GBM tissue correlated with favorable patient outcomes [91]. However, immune infiltration in pediatric low-grade glioma (pLGG) and pHGG has no bearing on survival, despite increased T-cell infiltration compared with normal brain tissue [92]. In DIPG, lymphocyte infiltration was not increased and the recruitment and activation of effector lymphocytes is rare. CD45 + CD3+ T lymphocytes in DIPG accounted for about 1.72% to 2.65% of the total CD45+ leukocytes, while GBM carried an abundance of infiltrating T lymphocytes, accounting for roughly 7.09% to 50.2%. Majzner et al. reported that some patients with mutant H3K27M harbored few infiltrating T-cells in their tumor tissue even after CAR-T treatment [93], suggesting a potential reason for DIPG resistance to ICIs.

### 3.4. Natural Killer Cells

Natural killer (NK) cells are cytotoxic effector cells in the innate immune system, and are essential for normal immune clearance [94,95]. The dysfunction of NK cells in tumors is strongly correlated with poor survival in patients with solid tumors, indicating its potential role in inhibiting tumorigenesis and tumor progression [96,97].

In adult GBM, NK cells can specifically target glioblastoma stem cells, and NKG2C+ NK cells are valuable for immunotherapy of glioblastoma [98,99,100,101]. In contrast to adult HGG, NK cell infiltration is absent in DIPG, and single-cell sequencing data suggest that only about 0.66% of lymphocytes are present in H3K27M-mutant DIPG patients [102]. Fortunately, the ability of NK cells to lyse DIPG cells was demonstrated when they were co-cultured in vitro [10], suggesting the potential significance of an anti-DIPG mechanism. However, clinical trials are still lacking to confirm the effectiveness of NK cells for DIPG.

### 3.5. Tumor Immune-Related Molecules

A lack of inflammatory mediators is an important feature of DIPG. DIPG-associated macrophages express markedly lower levels of IL6, IL1A, IL1B, CCL3, and CCL4, among other inflammatory factors than adult GBM-associated macrophages [9]. In vitro culture studies and RNA-sequencing of DIPG samples revealed that DIPG showed significantly reduced levels of both cytokines and chemokines, compared with adult GBM [77]. Immune escape mechanisms are dispensable for the maintenance of the DIPG immune microenvironment. The majority of infiltrating cells exhibit a loss of PD-1 or PD-L1 expression [92], and soluble NKG2D ligands are barely detectable in patients’ sera [102]. Only a few immunosuppressive factors, such as TGFB1, are detected in DIGP [9]. In general, immunosuppression is not a major component of the DIPG microenvironment when compared with adult GBM.

## 4. Immunotherapy for DIPG

The immune system adopts a series of complex mechanisms to detect and eradicate cancer cells [103]. These pathways could theoretically interfere with the progression of malignant tumors; however, surviving tumor cells after immune screening accelerate the disease process in cancer by avoiding the host’s anti-tumor immune response. Cancer immunotherapy was developed based on studies investigating the reactivation of anti-tumor immune responses to overcome immune escape-related pathways [7]. Currently, the main immunotherapies for DIPG include checkpoint inhibitors (ICIs), CAR-T Cells, vaccine therapy, and oncolytic viruses (Figure 3A).

### 4.1. Checkpoint Inhibitors

Programmed death 1 (PD-1) and cytotoxic T lymphocyte-associated protein 4 (CTLA-4) are immune checkpoint proteins that are involved in the initial suppression of T-cell function, playing a crucial role in tumor-mediated immunosuppression [104,105]. Checkpoint inhibitors block the inhibitory signaling when T-cells interact with tumor cells, by targeting PD-1 and/or CTLA-4 [106]. ICIs reactivate CD8+ T-cells in the tumor microenvironment and serve as an effective therapy in cancers, such as colon carcinoma, fibrosarcoma, and melanoma [107,108,109].

Unfortunately, patients with DIPG did not exhibit any survival benefit when they were treated with ICIs [11,12,13]. In a clinical trial, these patients that were treated with the PD-1 inhibitor, or pembrolizumab, manifested reduced median progression-free survival (PBTC045). This may be due to the rapid deterioration of the patient’s neurological function after the initiation of checkpoint blockade.

### 4.2. CAR-T Cells

The concept of CAR-T-cells was proposed in 1989 [110]. CARs have the ability to redirect T-cells to specific antigens. The activated T-cells that are infused into patients kill cells that are expressing specific antigens [111]. Considering the ineffectiveness of ICIs in DIPG therapy, CAR-T-cell therapy may represent a more appropriate treatment strategy. The specific recognition of an engineered antibody reflects the enormous significance of individualized and precise treatment [112]. CAR-T-cell therapy has shown encouraging success in hematological malignancies, such as B-cell acute lymphoblastic leukemia that was treated with anti-CD19 CAR-T-cell therapy, prompting investigation into other types of solid tumors [113,114,115].

However, despite sporadic preclinical studies and few case reports, evidence from large cohort clinical trials that suggests the effectiveness of targeting CAR-T-cells in glioma remains insufficient [116,117]. Currently, several DIPG antigens have been developed as the specific recognition antigens of CAR-T, such as IL13Rα2, GD2, EGFRvIII, and B7-H3 [118,119,120,121,122,123,124]. GD2 CAR-T-cell therapy is the most representative and efficacious intervention [93]. GD2, which is disialoganglioside, is highly expressed in DIPG tumor cells carrying H3.1K27M and H.3.3K27M, and its inhibition attenuates DIPG cell proliferation and tumorigenesis in vitro and in an animal intracranial model [120]. Therapies targeting GD2 are currently being investigated in various malignancies, including neuroblastoma, osteosarcoma, and melanoma [125,126,127,128]. In the latest clinical trial, it is encouraging that three patients that were diagnosed with H3K27M-mutant diffuse midline gliomas (DMG) showed clinical and imaging improvement after the first intravenous infusion of GD2 CAR-T-cells [93].

### 4.3. Vaccine Therapy

In recent years, cancer vaccines, as a new immunotherapy method, have gradually attracted the attention of researchers [129]. Cancer vaccine therapy emphasizes the fight against immune tolerance via the stimulation of foreign antigens (cancer-specific DNA, mRNA or polypeptide chains) to reactivate the immune system and induce an immune response against the tumor [130]. Cancer vaccines include several categories: genetic vaccines, tumor cells, immune cells, and peptides or proteins [131].

Peptide vaccines containing EphA2 and IL-13Ra2 antigens exhibited a specific immune response and advanced clinical survival of children with malignant brainstem and non-brainstem gliomas [132,133]. In addition, peptide vaccines that directly target H3K27M-specific proteins have been successful in preclinical studies of DIPG [134], and efficacious in clinical trials [135]. Further, autologous dendritic cell vaccines (ADCVs) also demonstrated negligible toxicity in normal cells and improved the clinical response [136]. A clinical trial evaluating ADCV is ongoing (NCT03396575).

### 4.4. Oncolytic Viruses

Oncolytic viruses are designed to target and kill cancer cells with minimal damage to normal cells [137]. Infection by the oncolytic virus itself and concomitant killing of cancer cells results in the release of large amounts of antigens, which elicit anti-tumor immunity and the activation of the tumor microenvironment. Oncolytic viruses can reprogram the immune microenvironmental phenotype, convert immune cold tumors to hot tumors, enhance the intrinsic capacity of immune cell infiltration, and stimulate cytokine release [138,139]. Essentially, this is a translational application of the concept of immune stimulation to immunotherapy [138,140].

Currently, two oncolytic viruses have been developed for the treatment of pediatric gliomas (containing DIPG). The first is adenovirus DNX-2401 (delta-24-RGD), initially demonstrating the safety and efficacy in an immunocompetent DIPG mouse model, by targeting and killing DIPG cells, as well as inducing an immune response [141,142]. A subsequent clinical trial (NCT03178032) further validated its therapeutic efficacy. Excitingly, in this clinical trial, an 8-year-old patient with DIPG demonstrated the utility of this oncolytic virus at the time of biopsy [143] (Table 2). Another oncolytic virus, herpes simplex virus 1716 (HSV1716), has been shown to exhibit potential anti-DIPG ability with an advantageous non-tumor tissue safety profile. HSV1716 greatly reduced the invasive capacity of DIPG cells and exhibited an exciting therapeutic efficacy in an orthotopic mouse tumorigenesis model [144].

## 5. Dilemma of Immunotherapy for DIPG

### 5.1. Non-Inflammatory Immune Microenvironment

As mentioned above, the sparse immune infiltration and insignificant host antitumor immune responses are the dominant roadblocks hindering immunotherapy in patients with DIPG. The negligible infiltration of antigen-presenting cells (APCs), such as DCs, B cells, and Th1 cells yields minimal concentrations of antigens that are presented to the lymphatic system, which is insufficient to activate the anti-tumor response of effector cells such as CD8^+^ T-cells and NK cells. Besides, the decreased secretion of immune-stimulating factors and pro-inflammatory cytokines is also one of the main factors contributing to the poor immune response (Figure 3A) [9,10,77]. The characterization of the DIPG immune microenvironment is still basically unknown, warranting further exploration. The resolution of the “cold” tumor immune microenvironment strongly favors immunotherapy.

### 5.2. Lower Mutational Load

The mutational burden of a tumor is defined as the number of mutations in each coding region of the tumor genome. For certain cancer types, a higher mutational load may be a significant predictor of enhanced clinical response to checkpoint inhibitors [145,146,147]. Although the efficacy of mutational burden in predicting response to treatment with ICIs is controversial [148], a lower response is still associated with a relatively lower mutational load in GBM [149]. Similar to adult GBM, a recent study reported that DIPG carries a lower tumor mutational burden that is associated with a lower level of neoantigens that are generated [150]. Therefore, checkpoint inhibition may not be an effective therapeutic approach targeting pHGG in the absence of a high tumor mutational burden [151].

### 5.3. Antigen Insufficiency, Attenuation, and Escape

In adult GBM, EGFR amplification and mutations and the loss of PTEN are common molecular features that are associated with neoantigens [152]. Compared with adult GBM, mutations such as EGFR gene amplification or PTEN deletion are less common in pediatric HGG, suggesting that a few common antigens in adult GBM may not be used in DIPG-based CAR-T [153,154,155].

The recognition of specific tumor antigens is one of the most critical steps in CAR-T-cell therapy [115]. Taking CD19+ CAR-T treatment of leukemia as an example, a decrease in the expression of tumor antigens below the threshold that was detected by CAR-T-cells can lead to a decline in efficacy and tumor recurrence [114,156]. In the clinical trials of CAR-T for GBM treatment, targeting IL13-Rα2 has attracted great attention. Unfortunately, no IL13-Rα2 was detected in patients who have relapsed after 6 months treatment [116]. However, targeting an identified neoantigen has yielded favorable results in preclinical studies, whereas antigen evasion and attenuation disqualify it as a universal drug target. Moreover, despite the high affinity of HLA peptides for a wide variety of HLA molecules, the heterogeneity of HLA types renders HLA molecules untargetable in patients with DIPG. In addition, the design of CAR-T-cells targeting neoantigens is expensive and time-consuming, and dysfunctional in patients with defective T-cells owing to disease or previous treatments [157].

### 5.4. Toxicity

The growing exploitation and application of immunotherapy has highlighted the importance of the identification and management of its toxicity and side effects. [15]. Currently, studies investigating immunotherapy-related toxicity mainly focuses on ICI and CAR-T-cell therapies [158]. ICI therapy can reverse the inhibitory effect of tumor cells on T-cells in the tumor microenvironment. However, the reactivation of T-cell function leads to the upregulation of inflammatory factors, which facilitates a spectrum of immune-related adverse events (irAEs) [159]. The irAEs encompass organ toxicities involving skin [160], gastrointestinal tract [161], hepatitis [162], endocrinopathies [163], thyroid [163], pituitary [164], pneumonitis [165], and rheumatologic manifestations [166]. Adoptive cell therapies, such as CAR-T-cell, can lead to the occurrence of cytokine release syndrome (CRS) and immune effector cell-associated neurotoxicity syndrome (ICANS) [167,168]. CRS is the most common side effect of CAR-T-cell therapy, characterized by the massive release of pro-inflammatory cytokines (IL-6, IFN-γ, and TNF-α). The inflammatory storm that is triggered by CRS leads to systemic inflammation and even multi-organ failure [169]. CRS occurs more frequently in children than in adults that are treated with CAR-T-cells, which requires careful evaluation of potential toxicity when administering CAR-T-cells to pediatric patients with DIPG [170]. ICANS is another potential side effect of CAR-T therapy, occurring in 40% of CAR-T-treated patients [171]. These toxicities impair the efficacy of immunotherapy and warrant specific management [158].

The immaturity and inherent fragility of the pediatric immune system poses a huge challenge to the immunotherapy of DIPG [172]. Schuelke et al. established four DIPG immunotherapy models to monitor brainstem toxicity that was induced by CAR-T-cell therapy, HSVtk/GCV suicide gene therapy, oncolytic virus therapy, and adoptive T-cell transfer. The results showed that all animal models that were treated with immunotherapy exhibited brainstem inflammation [173]. In a clinical trial of GD2-CAR T-cell therapy for H3K27M-mutated diffuse midline gliomas (DMG), an H3K27M+ spinal cord DMG patient exhibited Grade-3 CRS and Grade-4 ICANS [93].

### 5.5. Blood-Brain Barrier (BBB)

Under physiological conditions, T lymphocytes cannot cross the blood-brain barrier, but the presence of lymphocytes can be detected in the cerebrospinal fluid, suggesting that immune cells may enter the brain lesions through the cerebrospinal fluid and choroid plexus pathways [174,175]. DIPG manifests reduced permeability of the blood-brain barrier compared with the other pediatric gliomas that are located in the cerebral cortex [14].

Due to the compactness of the BBB in DIPG, fewer APCs infiltrate the brain parenchyma, which leads to the influx of tumor antigens into the lymphatic system at an insufficient concentration to induce an effective immune response against DIPG tumor cells [176,177,178]. In addition, the existence of the BBB leads to a huge discrepancy between the real therapeutic concentration of the lesion and the ideal hypothetical concentration of chemotherapy drugs, ICI, or infusion of CAR-T-cells and other therapies.

### 5.6. Cortisol Treatment

Dexamethasone is the most common initial treatment for newly diagnosed (or suspected) DIPG for the alleviation of neurological symptoms that are caused by the tumor [2]. It can effectively reduce the degree of edema and inhibit the inflammatory response [179]. As corticosteroids inhibit the permeability of the BBB, it may limit the infiltration of APCs and effector cells such as NK and CD8 + T-cells [180,181,182]. Besides, dexamethasone can upregulate CTLA-4, resulting in immunosuppression and the arrest of T-cells in the cell cycle [183]. In a study using an H3.3K27M-specific vaccine in DIPG, patients that were treated with dexamethasone showed higher levels of baseline circulating MDSCs, which were associated with a poor immunotherapy response and poor prognosis [135].

## 6. Immunogenic Cell Death (ICD)

Due to the low response to ICIs and immune-related adverse side effects that were mentioned above, a new immunotherapy concept is needed. ICD can reverse the tumor immunosuppressive microenvironment by boosting the efficiency of immunotherapy. ICD is a form of programed cell death that is induced by multiple antitumor therapies accompanied by the release of DAMPs, which facilitate the maturation of DCs and tumor death by CTLs [16]. The DAMPs that are exposed on, secreted from, or passively released by dying tumor cells can interact with phagocytotic, purinergic, or pattern-recognition receptors (PRRs) to activate the tumor immune microenvironment and ultimately boost antitumor innate and the adaptive immune response [184]. Mice showed resistance to challenges with live cancer cell lines after vaccination with cancer cell lines exhibiting ICD that was induced by anthracyclines, oxaliplatin, photodynamic therapy (PDT), or γ-irradiation in vitro [185]. Recently, several clinical trials have confirmed the clinical benefit of doxorubicin chemotherapy (a typical ICD inducer) combined with ICIs [186,187,188]. Patients with B-cell lymphoma manifested an improved clinical outcome when they were vaccinated with autologous tumor cells showing immunogenic death [189]. Moreover, modifying the delivery of ICD inducer formulations can further improve tumor regression rates and reduce adverse toxic effects. For example, cancer-activated doxorubicin prodrug nanoparticles (CAP-NPs) can specifically release cytotoxic doxorubicin when targeting cathepsin B-overexpressing cancer cells, which significantly reduces systemic adverse effects as well as greatly increases the rate of tumor regression in combination with ICIs [190]. Hence, ICD is a promising method to potentiate the immunotherapeutic efficacy synergistically.

### 6.1. Induction of ICD

ICD can be triggered by multiple stimuli, including but not limited to intracellular pathogens, conventional chemotherapeutics, and physical stress. The rapid induction of ICD contributes to the anticancer effect of many chemotherapeutics and physical therapies [191,192,193]. The inducers of ICD can be divided into Type I and Type II categories, according to different ICD inductive mechanisms. Type I inducers trigger ICD by promoting secondary or collateral endoplasmic reticulum (ER) stress including cardiac glycoside (CG), cyclophosphamide (CTX), doxorubicin (DOXO), mitoxantrone (MTX), oxaliplatin (OXP), ultraviolet C radiation (UVC), γ-irradiation (γ-IRR), and bortezomib [184]. By contrast, Type II ICD inducers are focused on ROS-based ER stress as in hypericin-based photodynamic therapy (Hyp-PDT) and coxsackievirus B3 (CVB3) infection [193]. The basic mechanism underlying ICD is ER stress induction and ROS production. The ability to induce ICD is largely determined by the degree of ER stress. Compared with secondary or collateral ER stress, focused ER stress is more immunogenic and releases an abundance of DAMPs frequently and effectively [194]. Therefore, Type I ICD inducers are preferred over Type II ICD inducers in terms of ICD-inducing capability. However, a substantial proportion of ICD inducers do not directly target ER but target the plasma membrane (channels or proteins), nucleus (DNA replication proteins), or cytosol [184]. Shikonin targets tumor-specific pyruvate kinase-M2 protein in the cytosol [195]. Cyclophosphamide targets DNA in the nucleus [196]. They induce ER stress and release of DAMPs via secondary or collateral effects. A small proportion of ICD inducers target ER stress. For instance, Hyp-PDT induces focused ROS-based ER stress owing to the ER-localizing ability of hypericin to release massive ROS at the ER when it is excited by light at a specific wavelength [193,197]. Hyp-PDT promotes protective antitumor immunotherapy via ICD-mediated release of DAMPs.

### 6.2. Emission of DAMPs

DAMPs play a critical role in ICD-mediated antitumor immune response. DAMPs act as a danger signal or adjuvant and can be exposed on the surface, passively released, or actively secreted by dying cells during ICD. ICD-associated DAMPs mainly contain high mobility group protein B1 (HMGB1), the small metabolite ATP, mtDNA, calreticulin (CRT), heat shock protein 90 (HSP90), Type I interferons (IFNs), and IL-1 family cytokines [198]. DAMPs can be classified into constitutive DAMPs that are inherently expressed in the cell, while inducible DAMPs are generated during ICD [199]. ICD-associated DAMPs display four significant characteristics. First, the production and emission mechanisms of DAMPs are dynamic in a defined spatiotemporal sequence, which depends on the death stage and stimulus type. DAMPs can be transferred to extracellular space via the classical secretory pathway and exocytosis in the case of cells in the pre-apoptotic stage [200]. The early-apoptosis secretion and exposure of DAMPs depend on autophagy and association with phosphatidylserine exposure, respectively [201]. In the middle or late stages of apoptosis, the defective plasma membrane contributes to the passive release of DAMPs [202]. Second, the number of DAMPs is closely associated with the type of stimulus that induces ICD. Doxorubicin promoted the surface exposure of HSP70, HSP90, and CRT, secretion of ATP and HMGB1, and upregulation of Type I interferons based on lipid peroxidation signaling-induced ER stress [190]. By contrast, colchicine induced the release of HSP70, HSP90, and HMGB1 [203]. Third, DAMPs exert biological effects mainly through binding pattern-recognition receptors (PRRs) that are expressed on the surface of DC and other immune cells. Fourth, DAMPs may be multifaceted contributing to tumor progression or metastasis depending on the extracellular microenvironment. Recent studies reported that some DAMPs such as HMGB1 and ATP accelerate tumor progression and contribute to resistance to anti-cancer therapies [204].

#### 6.2.1. HMGB1

HMGB1 participates in DNA repair and transcription as well as nucleosome stabilization, in addition to playing a role in immune regulation in the extracellular matrix [199]. The majority of anti-cancer agents and inducers of apoptosis and ICD such as doxorubicin, cardiac glycosides, septacidin, and Coxsackievirus B3, promote HMGB1 release [205]. HMGB1 release occurs in the middle and late phase of cell death via autophagy and plasma membrane defects after ICD induction. The priming of T-cells was inhibited in mice bearing CT26 cancer cells, owing to the depletion of HMGB1 or co-injection of anti-HMGB1 antibody [206]. HMGB1 boosted the antitumor immune response by binding TLR2 and TLR4 [207]. HMGB1 is as a strong cytokine, promoting the recruitment of various immune cells and the production of pro-inflammatory factors [208]. In addition, HMGB1 promotes DC maturation to potentiate antigen presentation and adaptive immune response [206]. A loss-of-function single-nucleotide polymorphism in breast cancer patients is associated with early relapse after anthracycline treatment [209].

However, HMGB1 may promote tumor invasion, while metastasis reduces anti-tumor immunity by interacting with the receptor for advanced glycation end products (RAGE) and T-cell immunoglobulin domain and mucin domain 3 (TIM3) [199]. The different behaviors of HMGB1 can be partially explained by the switch in redox states. HMGB1 is an inactive DAMP when fully oxidized. HMGB1 attracts immune cells in the fully reduced form.

#### 6.2.2. CRT

Calreticulin (CRT) can be surface exposed during the whole course of ICD (pre-/mid/late apoptosis) via different emission mechanisms [184]. CRT that is exposed on the surface of tumor cells experiencing ICD is an important “eat me” signal to promote tumor cell phagocytosis and antigen presentation via CD91, also known as LDL-receptor-related protein 1 (LRP1). The knockdown of CRT reduced the immunogenicity of tumor cells and suppressed the engulfment of anthracyclin-treated tumor cells by DCs in mice [210]. Further, recombinant CRT fragment 39-272 induced the activation and Ig class switching of B-cells [211].

#### 6.2.3. ATP

Similar to CRT, ATP was released in different stages of cell death/injury/stress [199]. ATP can be actively secreted before apoptosis, autophagy-dependent secretion in the middle of apoptosis, or passively released in the late stage of cell death. The function of ATP is related to its concentration in the extracellular space. Immature DC cells significantly increased their expression of MHC and costimulatory molecules (CD83, CD86, and CD54) after incubation for 24 h with 250 μM ATP. On the other hand, their ability to initiate T-helper 1 (Th1) responses was impaired [212]. Nevertheless, stimulation for less than 30min with 5mM ATP assists the maturation of human DCs and the release of IL-1β [213]. Stimulation of X2Y2 receptors on macrophages requires an EC_50_ < 1 μM ATP. An appropriate extracellular ATP gradient is established during chemotaxis of numerous immune cells. However, at the dose of 1 mM, ATP enhances immunosuppression by Tregs by activating P2Y2 receptor [204]. Extracellular ATP can be hydrolyzed to adenosine, a prominent immunosuppressive agent [214]. Therefore, ATP may also attenuate the anti-tumor immune response via immune suppression.

## 7. The Diagnostic and Therapeutic Potential of Induced ICD in DIPG

### 7.1. Advances in DIPG Immunotherapy

The discovery of several neoantigens, such as EZH2, WT1, B7-H3, HLA-A*02.01+ H3.3K27M26-35, and LSD1 provides the basis for the clinical transformation of immunotherapy [52,57,58,124,135,215]. In adult GBM, NK cells can circumvent the antigen loss that is seen in CAR-T-based models [152].

As mentioned above, GD2 CAR-T-cell and oncolytic viruses such as Ad-CD40L, DNX-2401, and HSV1716 show excellent safety and feasibility in the treatment of DIPG animal models [93,143,144,216]. However, how to deliver CAR-T-cells or oncolytic viruses to the brainstem of children with DIPG is still a challenge. Intraventricular infusion (ICV) is currently the predominant option. In a clinical trial of GD2 CAR-T for the treatment of patients with DIPG, ICV administration is associated with less systemic toxicity, reinforced secretion of pro-inflammatory cytokines, and minimal immunosuppressive cell populations that were detected in CSF, compared with intravenous infusion [93]. Moreover, clinical trials of intraventricular infusions of NK cells in pediatric patients with posterior fossa malignancies are underway, indicating excellent safety profile [217]. Chastkofsky et al. designed an oncolytic virus with an efficient delivery system via intra-tumoral injection of mesenchymal stem cells (MSCs) encapsulating the oncolytic adenovirus (CRAd.S.pK7) [218]. Several techniques are currently attempting to overcome the impediment of the blood-brain barrier, including the application of focus ultrasound [219], nanotechnology [220], and convection-enhanced delivery (CED) [221,222].

Currently, methods for assessing the efficacy of immunotherapy in vitro are aimed at directly measuring markers of tumor cell injury and death. Molecules that are released by dead cells, such as ATP, proteases, or lactate dehydrogenase (LDHA) are the mainstream indicators for evaluating curative effects in vitro [28,223,224]. Stallard et al. proposed that CSF circulating tumor DNA (ctDNA) can be used to quantify tumor growth and predict the treatment response [225]. Interestingly, CSF ctDNA was elevated during the treatment of patients with DIPG using GD2 CAR-T-cells [93]. In another clinical trial assessing H3.3K27M-specific vaccine, CSF ctDNA showed no apparent prognostic ability [135]. Surprisingly, lower levels of circulating MDSC (CD33 + CD11b + CD14+ HLA-DR^low^) are a prognostic indicator of DIPG and are associated with higher CD8+ T-cell response, prolonged survival, and higher vaccine efficacy, whereas Tregs did not show similar prognostic value [135].

### 7.2. Immunotherapeutic Modalities for Inducing ICD in DIPG

Based on the foregoing discussion, intratumoral delivery of vaccines, oncolytic viruses, CAR-cells, or chemotherapeutic drugs can induce the release of DAMPs in a potential therapeutic strategy, and the evaluation of ICD-related molecules in serum and cerebrospinal fluid may have potential diagnostic significance. Treatment combining ICD induction and immunotherapy is more efficient than immunotherapy alone in improving clinical outcomes and extending the overall survival. Adjuvant treatments that are mediated by ICD include radiation, vaccination with cancer cells experiencing ICD, oncolytic virus, and chemotherapy (Figure 3B) [226,227,228,229,230].

#### 7.2.1. Induction of ICD in Adult GBM

In adult GBM, immunocompetent mice that were vaccinated with early ferroptotic cancer cells that were induced by GPX4 inhibition exhibited more efficient phagocytosis and maturation of bone-marrow-derived dendritic cells (BMDCs), via the increased release of HMGB1 and ATP [231]. Further, the vaccination-like effect was defective in Rag-2^−/−^ mice, suggesting that the enhanced anti-tumor adaptive immune response was caused by early ferroptotic cancer cells. In a preclinical study of mice carrying a murine glioma cell line GL261-luc, Lim reported that the combination therapy of anti-TIM-3 antibody and stereotactic radiosurgery was more effective than anti-TIM-3 antibody alone in the regression of murine gliomas, accompanied with an elevated immune response [232]. However, they may have overlooked the possibility that luciferase-modified cells may themselves be immunogenic. The co-administration of anti-PD-1 antibody therapy with Zika virus (ZIKV) prolonged the overall survival of mice bearing primary tumors, owing to the oncolytic effect that was mediated by increased recruitment of CD8+ T and myeloid cells to brain tumor microenvironment [233]. Oncolytic adenovirus that was loaded with IL-7 combined with B7H3-targeted CAR-T therapy enhanced T-cell proliferation, increased tumor-infiltrating B7H3-CAR-T-cells, and extended the survival of mice bearing GL161 tumours, despite failing to reverse the exhaustion of B7H3-CAR-T-cells [234]. Kramer-Marek conducted photoimmunotherapy by conjugating an EGFR-specific affibody molecule to IR700, which generated ROS; induced ICD with the release of HMGB1, CRT, ATP, and HSP70/90; and promoted an anti-tumor immune response in a murine model [235]. Chemotherapeutic drugs, such as cyclophosphamide and bortezomib, combined with immunotherapy improved the overall survival by stimulating the immunogenic response of the brain tumor and anti-tumor immune response in animal models [229,236].

Further, novel biomaterials such as nanoparticles and hydrogels represent emerging and promising methods to facilitate immunotherapy by precisely targeting tumors and inducing ICD. An injectable hydrogel system was developed to induce ICD and suppress chemotactic CXC chemokine ligand 10 and indoleamine 2,3-dioxygenase-1, which significantly activated tumoricidal immunity after GBM surgical resection and attenuated the risk of relapse [237]. Nanoscale immunoconjugates with covalently attached antibodies to CTLA-4 or PD-1 successfully penetrate the BBB and stimulate tumoricidal immunity. Mice with intracranial GL261 GBM carried an increased number of CD8+ T-cells, NK cells, and macrophages as well as a decreased number of Tregs in the local tumor area after nanoscale immunoconjugate treatment [238]. Zhong et al. reported ApoE-mediated nano-delivery of granzyme B and CpG to induce ICD and enhance the anti-tumor immune response [239].

#### 7.2.2. Induction of ICD in DIPG

A retrospective cohort study reported that patients with recurrent and diffuse intrinsic pontine glioma that were exposed to repeated irradiation showed extended overall survival compared with those that were not exposed to repeated irradiation [240]. The benefit of re-irradiation may be partially explained by its role as an important ICD inducer in cancer cells. Besides, the efficacy and safety of autologous DC vaccines containing allogeneic tumor cell-line lysate was confirmed in patients with newly diagnosed DIPG. It evoked a specific anti-tumor immune response and ICD in T lymphocytes that were obtained from the cerebrospinal fluid (CSF) and peripheral blood mononuclear cells in the peripheral blood of patients [136].

An oncolytic adenovirus, termed Delta-24-ACT, was designed by Alonso et al. to express the costimulatory ligand 4-1BBL, which elevates the number and function of immune cells in the local tumor area. The administration of Delta-24-ACT in mice bearing DIPG orthotopic tumors remarkably improved the overall survival and boosted the long-term immunological memory against these tumors [241]. The treatment of adenoviruses expressing thymidine kinase (TK) and fms-like tyrosine kinase 3 ligand (Flt3L) in a mouse model of mACVR1 brainstem glioma increased the median survival by promoting recruitment of tumor antigen-specific T-cells [17]. (Table 3).

## 8. Conclusions

DIPG is the most lethal tumor involving the pediatric central nervous system. Immunotherapies such as ICIs, CAR-T-cells, vaccines, and oncolytic viruses present exciting opportunities to cure DIPG with unique and heterogeneous features. Nevertheless, the inadequate exploration of the DIPG immune microenvironment is currently a challenge that is hindering the application of immunotherapeutic modalities. DIPG is characterized by reduced infiltration of immune cells, decreased secretion of inflammatory factors, rare APCs, isolated and defective immune killing mechanisms, and a nearly deserted immune microenvironment. The non-inflammatory DIPG microenvironment greatly limits the development of immunotherapy for DIPG. Encouragingly, the induction of ICD, accompanied by the release of DAMPs demonstrates satisfactory efficacy of immune stimulation and antitumor response. Additional safety and efficacy studies using immunotherapies with ICD adjuvants are needed to treat DIPG.

## Figures and Tables

**Figure 1 pharmaceutics-14-01762-f001:**
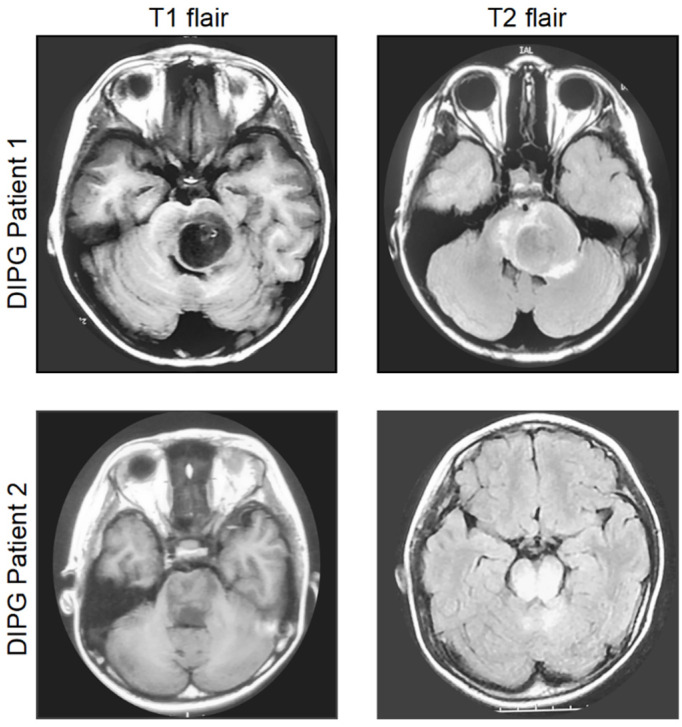
MRI images of two patients with radiographically classic DIPG. MRI image of two DIPG cases shows a space-occupying lesion in the pontine region, being relatively hypointense or isointense on T1-weighted images and hyperintense on T2-weighted images, when compared with a normal brain. DIPG Patient 1: 5 years, male; DIPG Patient 2: 7 years, female.

**Figure 2 pharmaceutics-14-01762-f002:**
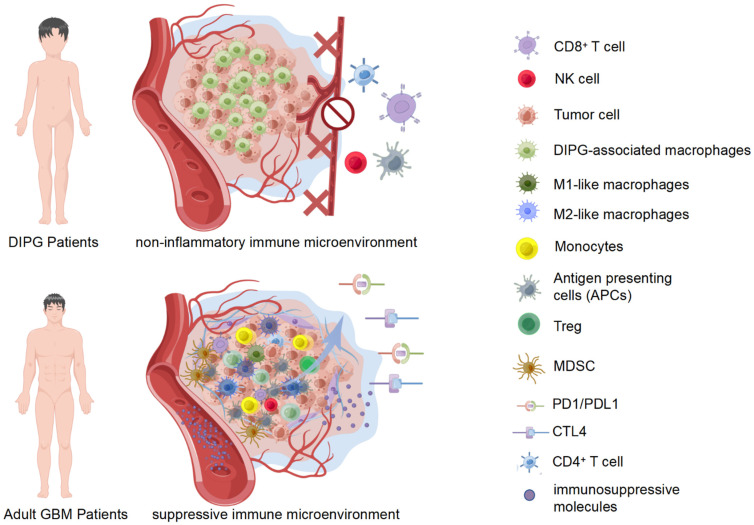
Characterization of the DIPG-associated immune microenvironment. DIPG is a “cold tumor”, indicating reduced immune cells infiltration, lower secretion of inflammatory factors, rare antigen-presenting cells, isolated and defective immune death mechanisms, and a nearly deserted immune microenvironment. Besides, DIPG-associated macrophages do not fully fit the M1 or M2 classification. In general, immunosuppression is not a major feature of the DIPG microenvironment, compared with adult GBM.

**Figure 3 pharmaceutics-14-01762-f003:**
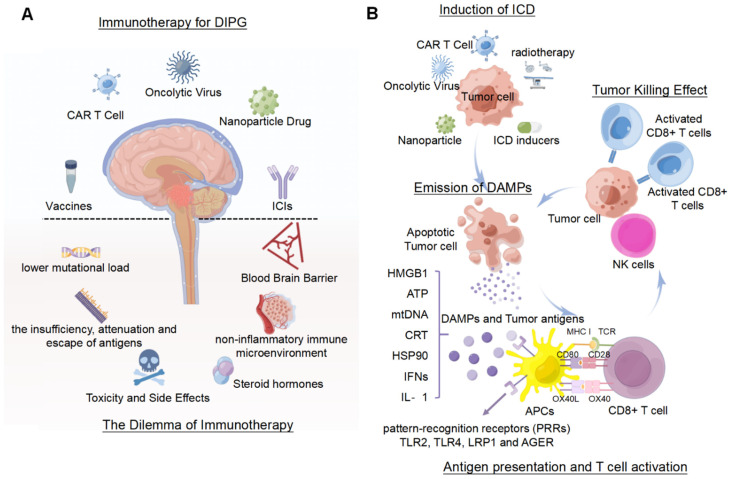
Types and dilemmas of immunotherapy for DIPG, and mechanisms by which immunogenic cell death enhances immunotherapy. (**A**) Cancer immunotherapy is based on the reactivation of anti-tumor immune responses and overcoming immune escape-related pathways. Current treatments utilize ICIs, CAR-T-cells, vaccines, and oncolytic viruses. The use of immunotherapy in DIPG is limited by a non-inflammatory immune microenvironment, lower mutational load, antigen insufficiency, attenuation and escape, toxicity of immunotherapy, blood brain barrier (BBB), and the use of cortisol. (**B**) ICD inducers can be combined with radiotherapy and immunotherapy for DIPG, resulting in cellular stress and cell death and the concomitant release of damage-associated molecular patterns (DAMPs), i.e., HMBG1, calreticulin, and ATP from dying tumor cells. Recruited dendritic cells assimilate the DIPG tumor antigens that are released from dying cells, such as HMGB1 binding to TLR2/4, which facilitates cytokine production and tumor antigen cross-presentation. The dendritic cells carrying DIPG antigens migrate to the cervical draining lymph node where they present the antigens to naive T-cells, thereby activating anti-DIPG effector T-cells. Primed effector T-cells migrate toward the tumor microenvironment and kill residual DIPG cells.

**Table 1 pharmaceutics-14-01762-t001:** Recurring genomic and proteomic alterations in DIPG.

Alteration	Mutated Categories	Prevalence	Description of Mutation	Refs.
H3.1K27M	HIST1H3B; Missense mutation	15%	H3 K27 trimethylation is ablated, generating an inhibition of the polycomb repressive complex 2 target genes, resulting in chromatin disaggregation and cellular aneuploidy.	[38,39,40,41]
H3.3K27M	H3F3A; Missense mutation	70%		
ACVR1	R206H, R258G, G328E/V/W and G356D; Mutation	25%	Encoding a serine/threonine kinase (ALK2) receptor with enhanced sensitivity to the ligand activin A, resulting in dysregulation of the BMP/SMAD pathway and increased tumor proliferation	[61,62,63]
TP53	G245S, R175H, R248Q, R248W, R273C, R273H, S241F, V157F; Mutation	75%	An increased co-occurrence with the H3 K27M mutation, increased DNA and protein instability resulting in decreased apoptosis	[64,65]
MYCN	Amplification	8%	DNA hypermethylation and chromosomal rearrangement leading to aneuploidy	[66,67]
ATRX	Depletion	10%	High co-occurrence with the H3 K27M mutation, causing destabilization of telomeres and altering gene expression in conjunction with the H3 K27M mutation	[45,68,69]
Receptor Tyrosine Kinase (RTK)	PDGFR, EGFR, FGFR; Amplification and mutation	60%	It occurs frequently with the H3 K27M-mutantion	[45,70,71]
Cell-cycle regulatory genes controlling RB phosphorylation	CDK4, CDK6, CCND1, CCND2, CCND3; Focal amplifications	30%	Inactivation of RB relieves negative regulation of the E2F transcription factor, permitting DNA synthesis and cell proliferation	[47,72]

**Table 2 pharmaceutics-14-01762-t002:** Current clinical trials of DIPG immunotherapy.

Type of Therapy	Treatment	Delivery Method	Patients	Efficacy	Ref. or NCT Number
ICIs	Ipilimumab/nivolumab	Convection enhanced delivery	Patients (*n* = 2)	Patient 1: dead; Patient2: progressive disease	[12]
	Pembrolizumab	Intravenous injection	Recruiting	Unknown	NCT02359565
	Pembrolizumab	Intravenous injection	Patients (*n* = 5)	Shorter median PFS than expected	PBTC045
	Pidilizumab	Intravenous injection	Active, not recruiting	Unknown	NCT01952769
CAR T-cell therapy	B7-H3-specific CAR T-cell locoregional therapy	Catheter into the ventricular system	Recruiting	Unknown	NCT04185038
	C7R-GD2 CAR T-cell therapy	Intravenous injection	Recruiting	Unknown	NCT04099797
	GD2 CAR T-cell therapy	Intravenous injection and intraventricular delivery	Patients (*n* = 3)	Patient 1: OS, 13 months; Patient 2: OS, 26 months Patient 3: OS, 20 months.	NCT04196413
Oncolytic virus	Oncolytic virus: DNX-2401	Intratumoral injection	Patients (*n* = 12)	Median OS: 17.8 months (range, 5.9 to 33.5)	NCT03178032
	Oncolytic virus: AloCELYVIR	Intravenous injection	Recruiting	Unknown	NCT04758533
	Oncolytic virus: Wild-type Reovirus + Sargramostim	Intravenous injection	Active, not recruiting	Unknown	NCT02444546
Vaccines	Peptide vaccine: SurVaxM	Subcutaneous injection	Recruiting	Unknown	NCT04978727
	Peptide vaccine: H3K27M peptide vaccine	Subcutaneous injection	Not yet recruiting	Unknown	NCT04808245
	DC vaccine: WT1 mRNA-loaded autologous monocyte-derived DCs	Intradermal vaccination	Recruiting	Unknown	NCT04911621
	DC vaccine: TTRNA-DCs	Intradermal vaccination	Recruiting	Unknown	NCT03396575

**Table 3 pharmaceutics-14-01762-t003:** Preclinical studies of induction of ICD to enhance the immunotherapy of glioma.

Treatment Category	Delivery Method	Model	Efficacy	Ref.
ICIs: antibody-PD-L1	Focused ultrasound combined with microbubble-mediated BBB opening (FUS-BBBO)	Mice/no glioma cells	NA	[242]
Early ferroptotic cancer cells	In vivo prophylactic tumor vaccination (Injected subcutaneously)	Mice/murine fibrosarcoma MCA205 or glioma GL261 cells	Attenuated the appearance of tumors at the challenge site	[231]
Anti-TIM-3 antibody + stereotactic radiosurgery (SRS)	Injected intraperitoneally /Stereotactic radiation	Mice/murine glioma cell line GL261-luc2	Long-term survival	[232]
Zika virus (ZIKV)	Stereotactic injection	Mice/GL261 or CT2A cells	Long-term survival	[233]
An interleukin-7-loaded oncolytic adenovirus (oAD-IL7) and a B7H3-targeted CAR-T	Stereotactic injection	Mice/GBM-Luc cells	Prolonged survival and reduced tumor burden.	[234]
EGFR-mediated photoimmunotherapy	Light exposure	Mice/GBM cells	Extensive tumor necrosis	[235]
Bortezomib and an oncolytic herpes simplex virus-1 (oHSV)	Intraperitoneally injected/intratumorally administrated	Mice/CAL27 cells	Prolonging survival enhance NK cell immunotherapy	[236]
An injectable hydrogel system	Intraperitoneally injected/intratumorally administrated	Mice/GL261	Suppressed tumor recurrence and prolonged the survival	[237]
Nanoscale immunoconjugates covalently attached antibodies to CTLA-4 or PD-1	Tail vein injection	Mice/GL261	Longer survival	[238]
ApoE peptide-functionalized polymersomes encapsulating granzyme B (ApoE-PS-GrB)	Tail vein injection	Mice/LCPN cells	Delayed tumor progression and prolonged survival time	[239]
Oncolytic adenovirus: Delta-24-ACT	Intra-tumoral injection	BALB/c mice/murine NP53 and XFM cell lines	Long-term survivors that developed immunological memory against DIPG	[241]
Adenoviruses expressing thymidine kinase (TK) and fms-like tyrosine kinase 3 ligand (Flt3L)	Intra-tumoral injection	Mice/ brainstem glioma harboring mACVR1	Recruitment of antitumor-specific T cells, and increased median survival	[17]

## Data Availability

Not applicable.

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
