# Peer review of "Immunogenic Cell Death Enhances Immunotherapy of Diffuse Intrinsic Pontine Glioma: From Preclinical to Clinical Studies"

_pharmaceutics, 2022, doi:10.3390/pharmaceutics14091762_

Round 1

Reviewer 1 Report

Major

1. Starting by the title: some clue that important part of the data is coming from preclinical studies is needed (e.g. insights from preclinical and clinical studies). 
2. This brings us to the next point. Authors, probably in an unintentional way, mix in the same section preclinical and clinical studies. And this should be fixed and clearly stated when it rises from preclinical, clinical or either in vitro cultured cells, if it was an immunocompetent or immunodeprived model and more details. We are dealing with factors which may need either immunocompetent models, or detailing how immunodeprived models covered the immune-related issues.

3. There is an overuse of the "etc" along the whole text. Please be specific whenever possible, at most and without overusing it, you can use "amongst others". And of course, DO NOT use it instead of "et al", please. Moreover, a great amount of sentences start with "And". Note that and is a connection, so either joint sentences, or make two different sentences but disregard the "and". 

4. Extensive revision by a native English speaker is needed. There are huge amounts of weird, unusual, confuse or directly wrong sentences. A new revision of the corrected/improved version is needed before any acceptance. Some examples below, only as an example (much more along the text)

- In the DIPG immune microenvironment, M1-like macrophages (CD68+CD163+) were not significantly different in DIPG, but CD45+Iba1+ microglia were increased compared with normal brain tissueIn vitro co-culture system, H3.3-K27M DIPG cells manifested the deficiency of repolarizing macrophages. Confusingly, the proportion of CD45+CD11b+ macrophages in CD45+ leukocytes were 95%, and that in adult GBM was 70%. It is marvelous and worthy to explore that the discrepancy between DIPG’s high isolation and microglia’s high activation (lines 170-176, needs rephrasing and more clarity, it is confusing and hard to read)

-Unlike the linear effect of the amount of TIL infiltrated, the phenotype of that also determine clinical outcome and therapeutic effect (lines 179-180, it is confusing and seems that something is missing)

And DIPG-associated macrophages expressing markedly lower levels of IL6, IL1A, IL1B, CCL3, CCL4, among other inflammatory factors (line 211-213, incomplete sentence) 

The dysfunction of NK cell mediated by tumor cell, is highly correlates with poor survival in patients with solid tumors, indicating the potential function of its in inhibiting tumorigenesis and tumor progression (line 199-201, basic English mistakes)

Programmed death 1 (PD-1) and cytotoxic T lymphocyte-associated protein 4 (CTLA-4), as immune checkpoint proteins that involved in the initiation of T cell functional suppression, play a crucial role in tumor-mediated immunosuppression (line 253-255, please revise grammar)

-In vaccine therapy for DIPG, peptide vaccines containing EphA2, IL-13Ra2 antigens has been exhibited specific immune response and advances in clinical survival (line 295, please revise grammar)

Taking CD19+ CAR-T treatment of leukemia as an example, once the expression of tumor antigens is lower than the threshold that can be detected by CAR-T cells, leading to the decline of efficacy and tumor recurrence (353-354 line, revise grammar)

-The inducers of ICD can be divided to type I ICD inducers and type II ICD inducers according to different ICD inductive mechanisms (line 450. divided into)

The resistance to rechallenge with live CT26 cancer cells was comprised in mice immunization with HMGB1 depletes CT26 cancer cells or co-injection of anti-HMGB1 antibody (lines 505-506, I don't even understand this sentence sorry, needs rephrasing)

Stefanie Stallard etc. proposed that CSF circulating tumor DNA (ctDNA) in have the power to quantify tumor growth and predict the treatment response (line 568-569, please don't use etc and revise grammar)

Surprisingly, lower circulating MDSC (CD33+CD11b+CD14+ HLA-DRlow) could be a prognostic indicator in DIPG patients, and it was associated with higher CD8+ T Cell responses, prolonged survival, and higher lower vaccine efficacy, but Tregs (another immunosuppressive associated cell) did not have no similar prognostic power (line 573 to 575 is it higher or lower vaccine efficacy? clarify)

Taken above description together, intratumoral delivery of vaccines, oncolytic viruses, CAR-T cells, or chemotherapeutics that could induce the release of DAMPs may a potential therapeutic strategy (lines 579-580, word missing?)

When anti-PD-1 antibody therapy co-administrated with Zika virus (ZIKV), the overall survival was elevated owning to the oncolytic effect mediated increased recruitment of CD8+ T and myeloid cells to brain tumor microenvironment (lines 596-597, rephrasing needed, please revised also grammar)

AND MUCH MORE. I have just stated some examples to help authors.

5. Clarification or correction needed in some points, see below

In a clinical trial, DMG patients treated with the PD-1 inhibitor, pembrolizumab, even experienced reduced median progression-free survival (line 262). Do they explain why? It would be good to state here

however, it has received great attention and supplementation in recent years, due to its enormous value of clinical transformation (line 266, this is a totally empty sentence without support. Still, I don't understand the word "supplementation" in this context

Encouragingly, CAR-T cell therapy was demonstrated as a promising efficacy in glioblastoma (LINE 276)

I don't really agree with it, CAR-T approaches are of limited use in several solid tumours. ref 104, by the way, is ONE CASE REPORT and 105 is incomplete I had to search for it to read the context.

Specifically, in ref 105 they said:

"Despite encouraging evidence of clinical safety and bioactivity for GBM-targeted CAR T cells, the overall response rates have been unsatisfyingly low, especially as compared to the remarkable clinical responses achieved against B cell malignancies"

Their results showed that they have increased survival in a group of n=6 xenografts. I'm sorry this do not really support that they are promising, at any point

In the dose of EC50 >100 μM, ATP is capable of activating P2X7 receptor on the dendritic cells to boost adaptive immune... (Line 534)

the reference cited is a review. Not acceptable. Authors MUST go to the original reference. which is  also a review, so what authors need is finally find the original reference with data

Michael Lim found that triple therapy of dual PD-1 and TIM-3 blockade with radiation contributed to 100% overall survival in mice bearing murine glioma cell line GL261-luc2, significantly higher than double or single treatment (line 593-594)

authors must be more specific. Please do not devote to simply copy paper sentences.

Mice treated with triple therapy survived 100 days what they called "long term survival". Moreover, authors may recognize that as was already stated, modification of cells with luciferase may be immunogenic by itself producing overoptimistic results https://www.liebertpub.com/doi/10.1089/hum.2014.048

Gabriela Kramer-Marek investigated a photoimmunotherapy by conjugating EGFR-specific affibody molecule to IR700, which can produce ROS, stimulate ICD with the release of HMGB1, CRT, ATP, and HSP70/90, and promote anti-tumor immune response after specifically defining the tumor margins (601-603) I don't understand why defining tumor margins is relevant here please explain

Chemotherapeutic drugs, such as cyclophosphamide and bortezomib, synergized with immune therapy improve overall survival through stimulating immunogenetic of brain tumor and anti-tumor immune response (605-606) please explain immunogenetic meaning in this context

Minor

1. Several abbreviations not defined in their first appearance

HGG (LINE 110) 

GBM (line 139) 

pLGG and pHGG (line 188)

DMG (line 285)

2. standardize immunotherapy vs immune therapy

3. Figures

Figure 1: please specify at least age of the patient and the magnetic field of the scanner used. Origin of the images needed if they are not produced by any of the authors

Figure 3A is fine but figure 3B lacks definition at some of the letters, and the fact that some text appear over parts of the figure hampers proper reading 

Still, I don't think "etc" is a good addition to a figure or figure legend. 

4. Maybe mutations could summarized in a table (page 3)

5. Nanoscale immunoconjugates with covalently attached a-CTLA-4 or a-PD-1 (line 613) please define a-CTLA-4 and a-PD-1

6. Reference sections have incomplete data in 40 (FORTY) references, see below

5

18

19

24

27

29

31

32

37

50

60

72

87

88

90

92

93

96

97

98

103

105

113

114

115

128

140

142

165

166

168

196

197

207

210

216

217

218

219

227

Author Response

Dear professor:

We really appreciate the time and effort you have spent reviewing our paper. We are pleased to note that you pointed out the comments, questions and suggestions to help us improve the quality and presentation our work (ID: pharmaceutics-1817524).

Motivated by your insightful comments, we deeply reconsidered and tried our best to address all problems your mentioned. We greatly thank you for all valuable comments provide on our submitting manuscript.

Again, many thanks for your patience and carefulness, and we are looking forward to hearing from you regarding our submission. We are ready to respond to any further questions you may raise.

Best wishes!

Reviewer 2 Report

Interesting topic with a comprehensive view of DIPG immune microenvironment. The text should be proofread for phrases that lack clarity i.e Adjuvant treatments mediated by ICD include radiation, vaccination with cancer cells experiencing ICD, oncolytic 585 virus, chemotherapy

Or for repetitive phrases that impede text understanding (line 457-458):

The common principle to evoke ICD is ER stress induction and ROS production.  The ability to induce ICD and immunogenicity of cell death is largely determined by the  focus on ER stress

The microenvironment of gliomas are highly different from systemic cancers. How is ICD expected to modify an immune microenvironment dominated by myeloid cells, noteworthy: microglia and tumor associated macrophages and MDSC? 

few corrections: CD11b+CD45+Iba1+ are not sufficient pour define a tumor associated macrophages. If you wish to maintain this designation, CD45hiCD11bIba1  is a more appropriate marker description. However, it is not coherent to use the cytometric markers only for this cell lineage without defining the other cells as well.  

Extensive english editing is imperative

Author Response

(The authors gave the same response as above.)

Round 2

Reviewer 1 Report

Dear authors,

Manuscript has clearly improved. Minor changes still needed please see below.

Minor comments

1.Title

Immunogenic cell death enhances immunotherapy of diffuse 2 intrinsic pontine glioma: from preclinical to clinical studies (remove "the")

2. Table 1

RTK mutation explanation: if the comment refers to RTK mutation, it OCCURS (not occur)

3. Line 162

Define SHH

4. Line 168

And the proportion of of macrophages in CD45+ leukocytes WAS 95% compared with 70% in adult GBM

(this 'was' refers to the proportion which is singular. the English has improved but it is clear that it was not applied to the new text, only prior text)

5. Line 169

Co-cultured with what? not clear

6. Line 267

"small" is not suitable in this context. What did authors mean?

few case reports?

7. Line 306

change the second "and" for "as well as"

8. Line 334

I suggest to change "disputed" for "controversial", a more standard word

9. Lines 514-516 (please modify accordingly see below)

Immature DC cells significantly increased their expression of MHC and costimulatory molecules (CD83, CD86, and CD54) after incubation for 24h with 250 μM ATP. On the other hand, their ability to initiate T-helper 1 (Th1) responses was impaired [212].

10.Line 540

Remove "the"

11.Text between lines 531-542 is weird at some points, please revise. 

12.Line 576

Change "but" for "however"

13.Line 583

Change "cell" for "tumours"

14.Line 587

Change "mice" for "a murine"

15.Line 590

"models", not "model"

16.Line 615

change "partly" for "partially"

17.Line 616

This sentence is a bit confusing 

"safety of autologous DC vaccines treated with allogeneic tumor"

it seems that vaccines are treated, so please revise the sentence.

Moreover, after several sentences about preclinical studies, this paragraph about human treatment comes totally out of the blue. Consider changing this sentence of place

18.Line 630

strange sentence

"immune death mechanisms"

19.References still have some problems

Revise

32, 50, 66, 72, 73, 83 , 94, 97, 98, 99, 112, 125, 131, 136, 137, 143, 144, 152, 171, 182, 190, 193, 199, 203, 219, 221, 231, 233, 239, 240, 242 

20.standardize whether page ranges are 1314-1319 or 1314-9

Author Response

Dear professor:

We really appreciate the time and effort you have spent reviewing our paper. We are pleased to note that you pointed out the comments, questions and suggestions to help us improve the quality and presentation of our work (ID: pharmaceutics-1817524).

Motivated by your insightful comments, we deeply reconsidered and tried our best to address all problems your mentioned. We greatly thank you for all valuable comments on our submitting manuscript.

Best wishes!

Reviewer 2 Report

The authors answered my questions 

Author Response

Dear professor:

We really appreciate the time and effort you have spent reviewing our paper.

Many thanks for your patience and carefulness.

Best wishes!